# A hybrid approach for intrusion detection in vehicular networks using feature selection and dimensionality reduction with optimized deep learning

Fayaz Hassan[1], Zafi Sherhan Syed[1], Aftab Ahmed Memon[1], Saad Said Alqahtany[2], Nadeem Ahmed[1], Mana Saleh Al Reshan[3,4], Yousef Asiri[5], Asadullah Shaikh[3,4]*

1 Department of Telecommunication Engineering, Mehran University of Engineering and Technology, Jamshoro, Pakistan, 2 Faculty of Computer and Information Systems, Islamic University of Madinah, Madinah, Saudi Arabia, 3 Department of Information Systems, College of Computer Science and Information Systems, Najran University, Najran, Saudi Arabia, 4 Emerging Technologies Research Lab (ETRL), College of Computer Science and Information Systems, Najran University, Najran, Saudi Arabia, 5 Department of Computer Science, College of Computer Science and Information Systems, Najran University, Najran, Saudi Arabia

* asshaikh@nu.edu.sa

**Data Availability Statement:** All relevant data are within the manuscript.

**Funding:** The author(s) received no specific funding for this work.

## Abstract

Autonomous transportation systems have the potential to greatly impact the way we travel. A vital aspect of these systems is their connectivity, facilitated by intelligent transport applications. However, the safety ensured by the vehicular network can be easily compromised by malicious traffic with the exponential growth of IoT devices. One aspect is malicious traffic identification in Vehicular networks. We proposed a hybrid approach uses automated feature engineering via correlation-based feature selection (CFS) and principal component analysis (PCA)-based dimensionality reduction to reduce feature matrix size before a series of dense layers are used for classification. The intended use of CFS and PCA in the machine learning pipeline serves two folds benefit, first is that the resultant feature matrix contains attributes that are most useful for recognizing malicious traffic, and second that after CFS and PCA, the feature matrix has a smaller dimensionality which in turn means that smaller number of weights need to be trained for the dense layers (connections are required for the dense layers) which resulting in smaller model size. Furthermore, we show the impact of post-training model weight quantization to further reduce the model size. Results demonstrate the effectiveness of feature engineering which improves the classification f1score from 96.48% to 98.43%. It also reduces the model size from 28.09 KB to 20.34 KB thus optimizing the model in terms of both classification performance and model size. Post-training quantization further optimizes the model size to 9 KB. The experimental results using CICIDS2017 dataset demonstrate that proposed hybrid model performs well not only in terms of classification performance but also yields trained models that have a low parameter count and model size. Thus, the proposed low-complexity models can be used for intrusion detection in VANET scenario.

**Competing interests:** The authors have declared that no competing interests exist.

## Introduction

The Intelligent transport system (ITS) can provide reliable and efficient communication service for intelligent devices and vehicular ad hoc networks. A key goal of ITS, as depicted in Fig 1, is to enable real-time interaction between vehicles, their drivers, roadside infrastructure like cloud servers, and pedestrians.

This level of connectivity is vital for autonomous and semi-autonomous vehicles to safely operate as part of an intelligent vehicular network. The ITS applications, utilizing vehicular network, are crucial for their operation. However, the connection provided by vehicular networks can also make them vulnerable to cyber-attacks, which have become increasingly prevalent with the growth of IoT devices. Cybersecurity has become a critical component to safeguard organizational boundaries, and identifying malicious traffic is a crucial research problem. Intrusion Detection Systems (IDS) are vital tools for detecting cyber attacks, but traditional rule-based systems struggle to keep up with the rapidly evolving threat. The

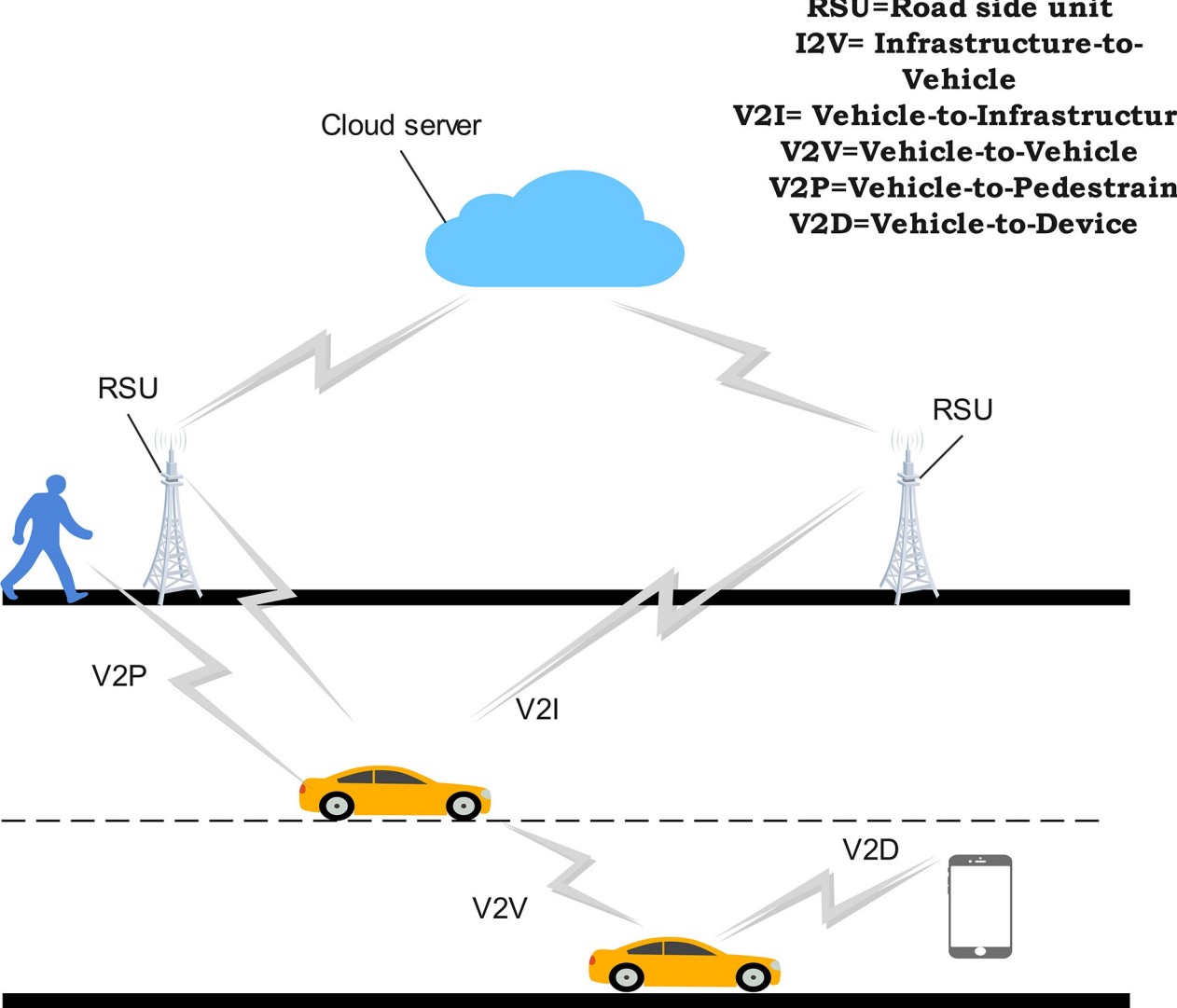

**Fig 1. Illustration of communication in vehicular network.**

widespread adoption of vehicular networks in ITS framework comes with notable challenges, particularly concerning data security [1]. Vehicular networks have the potential to transform transportation systems by enabling advanced communication between vehicles and infrastructure. Many research studies [2] have focused on enhancing the security of vehicular networks. To effectively address these security concerns, there is a need for network protection techniques that can detect and mitigate attacks. The rapid growth of vehicular network systems has revealed significant vulnerabilities, emphasizing the need for smart detection systems specifically designed for vehicular networks [3]. Traditional approaches for intrusion detection include the use of methods such as antivirus software, encryption-decryption, firewalls, and protocol control. While successful against limited attacks, these struggled with a high rate of false positives and numerous undetected attacks, notably complex Denial-of-Service (DoS) attacks [4]. Recent research shifted toward machine learning

(ML) techniques for intrusion detection, boosting identification rates, and managing large attacks more efficiently. However, machine learning algorithms are incapable of detecting intricate feature-integrated attacks and also perform poorly when noisy and multi-dimensional traffic data. Addressing these limitations, Deep Learning (DL) algorithms, like Deep Neural Networks (DNN) [5], Convolutional Neural Networks (CNN) [6], and Recurrent Neural Networks (RNN) [7], have emerged as advanced tools for intrusion detection. These nonlinear structured DL models have shown promise in improving accuracy and handling complex attack scenarios [8, 9]. In this study, our primary objective is to establish an ideal intrusion detection model with a high-performance model utilizing a two-stage deep learning algorithm. In the first stage, we employ FS before implementing the Neural Network (NN). This strategic placement of FS aims to effectively reduce the number of neurons, subsequently reducing model parameters. The second stage involves further size reduction through quantization. The initial stage capitalizes on statistical methods such as feature selection and dimensionality reduction. The second stage adopts a DL approach, Fully connected Neural Network (FCNN). To streamline model size, we leverage TensorFlow model optimization toolkit 1, utilizing its capabilities to minimize model dimensions. The main contributions of this study can be summarized as:

- We propose an efficient two-stage DL-based IDS framework for identifying Attack and Benign communication in the vehicular network. The designed model retains high detection rate for malicious data whilst also of low-complexity and reduced model size. Consequently, the proposed model has been tested on the CICIDS2017 dataset. To achieve comprehensive evaluation for the imbalanced label distribution, the model is evaluated is terms of metrics such as F1-score and loss.

- To reduce the number of parameters and size of the model in DNN, we implement the Correlation-based Feature Selection (CFS) technique to remove features which are highly correlated. As an alternate to CFS, Principal Component Analysis (PCA) is used that reduces the feature matrix dimensionality by first mapping the matrix in terms of eigen vectors, and then retaining vectors which describe most variance in the data.

- We propose the use of post-training model quantization technique to further reduce the model size and hence improve the computational.

This study is organized as follows: The section of *Related Work* presents a review of recent advances in intrusion detection. The *Proposed System Methodology* section summarizes the research gap and the problem statement followed by a detailed description of the proposed system methodology, including the dataset, data preprocessing steps, feature selection, dimensionality reduction, and the fully connected neural network (FCNN) architecture, as

well as post-training model quantization techniques. Experiment results as well as evaluation of the model's performance for both binary and multi-class classification tasks is provided in the section of *Experimental Results*, with an in-depth discussion of the findings provided in the *Discussion* section. Finally, conclusions and potential directions for future work are provided in *Conclusion and Future work* section.

## Related work

Many studies evaluating the deep learning intrusion detection models have used binary classification and accuracy performance criteria. The detection rate is unsatisfactory [10]. Using an old KDD intrusion detection dataset, several research studies assessed intrusion detection methods. The KDD99 dataset does not consider many recent cyberattacks, and the IoV network was not considered when developing the dataset. By using reinforcement learning to adapt evaluation strategies across different driving scenarios trust management model has been proposed [11] to evaluate the trustworthiness of received messages, ensuring that unlawful data does not impact driving decisions. Privacy Preserving Reputation Updating scheme has been discussed [12] for cloud-assisted vehicular networks to address the bottleneck issue in reputation management. This approach significantly reduces computation and communication overheads.

A two-stage IDS was proposed by Mushtaq et al. [13] by combining the Auto-Encoder and the LSTM. The suggested model achieved an accuracy of 89.00% for attack classification using the NSL-KDD dataset.

Lin Zhang et al. [14] developed CNN-based intrusion detection models and compared their performance against other conventional classifier. They reported that CNN models achieved highest classification performance. A similar trend was reported in [15, 16] where several types of conventional machine learning techniques, models, and methodologies were used to identify the network intrusion detection, however, these did not have the capabilities of the deep learning based models.

In [17], Alzharani and Hong suggest the integration of an Artificial Neural Network (ANN) employing a signature-based approach for identifying Distributed Denial of Service (DDoS) attacks in IDS. Results show that the combined strategy of the signature-based method and ANN was more accurate, attaining an impressive value of 99.98% Moreover, Kim et al. [18] designed the AI- IDS-based DNN model using real-time data and the publically available CICIDS 2017 dataset to evaluate the performance of deep learning models. The model accuracy was 98.07%. Also, discussed the limitation of present IDS detection because of the critical nature of security. The study identifies the use of a CNN-LSTM based classifier for for payload-level intrusion detection. An alternate solution based on Monte Carlo tree search method is proposed by Zhang et al. [19]. This approach relies on the use of adversarial instances of cross-site scripting attacks as part of the model training process. Additionally, the authors employed the Generative Adversarial Network (GAN) framework to improve the IDS capability to detect malicious attacks. Experimental evaluations were conducted using the CICIDS-2017 dataset to craft novel XSS attacks. The accuracy exceeding 99.90% to identify XSS attacks using GAN detection model. successfully An integrated deep learning model for anomaly identification in IoT networks was created by Yin et al. [20]. To find the anomalies, they used CNN and LSTM autoencoder [21, 22]. They improve learning predictions by using a two-stage window-based data preprocessing. Their proposed method, which was limited to binary classification, produced better precision, recall, accuracy, and F1-score results. In [8], Barik et al. trained an LSTM model with 128 input nodes, 3 hidden layers, a dense layer, and a drop of 0.2 for all hidden layers. The model was evaluated using Hogzilla and ISCX 2012 datasets,

for which the authors reported an accuracy of 98.88%. An AE-LSTM-based approach was proposed by Mahmoud et al. [23] to identify anomalies in an IoT environment. The NSL-KDD dataset was used to train the model, classifying attacks with an accuracy of 98.88%. No data pre-processing was done in the study to enhance training effectiveness. Another pertinent study involves the utilization of autoencoders for detecting anomalies, as explored in [24]. Here, autoencoders are employed to grasp the essence of a standard network profile via non-linear feature reduction. The research shows that the normal (non-malicious) instances within the test dataset exhibited minimal reconstruction errors during implementation, while anomalous entries within the same dataset resulted in substantial reconstruction errors. In [25] proposed the Modified Genetic Algorithm (MGA) and a Long Short-Term Memory (LSTM) network to detect cyberattacks in IoT networks, particularly in smart city environments. Their approach focuses on edge computing in centralized IDS, addressing challenges in real-time detection and feature optimization. In [26], ensemble learning techniques Voting Gray Wolf Optimizer (GWO) used to enhance the accuracy of Intrusion Detection Systems (IDS) for IoT networks. Another study [27] the author highlights the potential of hybrid techniques combining feature selection and optimization methods to improve IDS effectiveness.

Furthermore,in [28, 29], the author used IDS, which is based on deep learning approach to secure vehicular network. One finds that prior works use 'accuracy' as classification performance metric even though datasets have an imbalanced class distribution, which means that performance is always optimistic in favor of the majority class that is often the non-malicious normal traffic. This is a major weakness as it means that the results do not show the correct picture of the IDS performance.

Model optimization, in terms of classification performance and size, is not adequately addressed in existing research literature. One notes that in a majority of cases, model optimization is limited in scope to increasing classification performance and does not address reducing model size or reducing the number of parameters through feature engineering.

Our literature survey does not find prior work that studies the impact of using quantization for model size reduction on the classification performance for VANETs. The proposed addresses these three gaps in pursuit of developing a hybrid models for IDS for scenario.

## Proposed system methodology

The process flow diagram of our proposed methodology for hybrid IDS is illustrated in Fig 2. The proposed technique comprises three distinct steps. The initial step involves data pre-processing, which includes cleansing the data by removing 0s, nulls, and infinity values and scale any out-of-range instance found in the datasets. This step is crucial for ensuring data integrity for the IDS. Next, we experiment with two methods to reduce the size of feature matrix before it is fed to the classifier. The first approach uses feature selection whereas the second method uses dimensionality reduction using PCA. The next step involves the training and evaluation of FCNN that uses dense layers for classification. Finally, we leverage post-training model weight quantization to reduce the model size.

Details of each aspect of the system model are provided in the following sub-sections.

### Dataset

The Canadian Institute for Cybersecurity IDS (CICIDS) 2017 dataset [30] was used to evaluation our proposed framework. The dataset is provided by University of New Brunswick, Canada, as a set of CSV file in Packet capture(PCAP) format. The specifics of each file are read using the Pandas package and combined their data to yield a single dataframe. This dataframe has a shape of (2830743, 79), where the number of rows represent instances and the number of

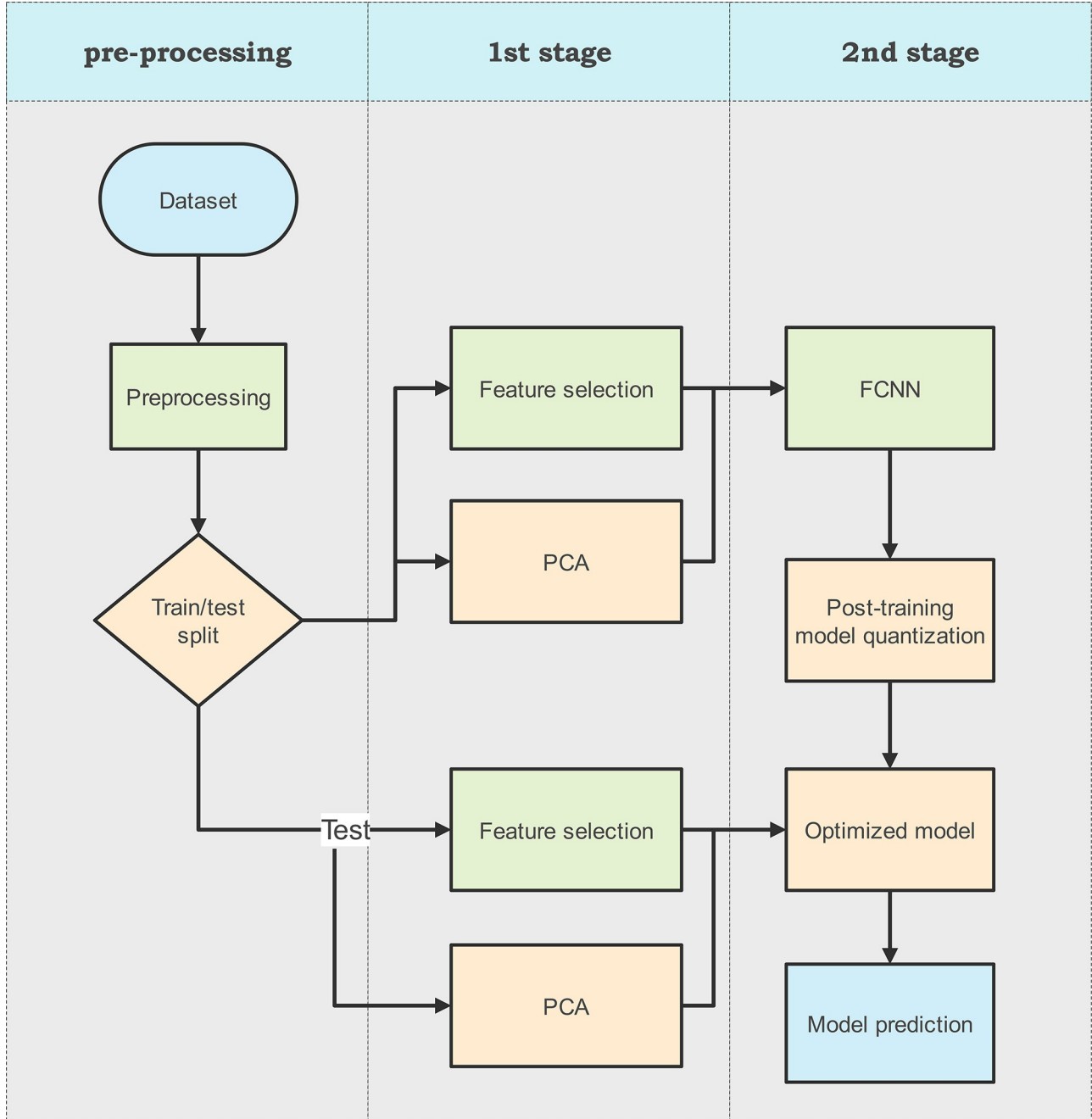

**Fig 2. Proposed framework.**

columns represent features or labels (the last column). The values in the Label column contains names of Attack or Benign communication. Since labels can be different for binary and multi-label classification, therefore, we renamed this column from "Label" to '"MultiClass Label" so that the column name reflects the nature of data in the column. Label distribution provided in Table 1, from where one notes that large imbalance between the occurrence of different events.

**Table 1. Summary of dataset records for each class.**

| Label | Number of records |
|---|---|
| Benign | 2273097 |
| Bot | 1966 |
| DDoS | 128027 |
| DoS GoldenEye | 10293 |
| DoS Hulk | 231073 |
| DoS Slowhttptest | 5499 |
| DoS slowloris | 5796 |
| FTP-Patator | 7938 |
| Heartbleed | 11 |
| Infiltration | 36 |
| Port Scan | 158930 |
| SSH-Patator | 5897 |
| Web Attack/SQL Injection | 21 |
| Web Attack/Brute Force | 1507 |
| Web Attack/XSS | 652 |

**Table 2. Summary of dataset records for each class.**

| Label | Number of records |
|---|---|
| Benign | 2273097 |
| DoS Hulk | 252661 |
| Port Scan | 158930 |
| DDoS | 128027 |
| Brute force | 13835 |
| Web Attack/Brute Force | 2180 |
| Bot | 1966 |

We then combined certain labels which belong to the same category on a macro-level. For example, we merged "DoS GoldenEye", "DoS Hulk", "DoS Slowhttptest", and "DoS slowloris" attacks under a new category of "DoS" attacks. Similarly, we combined the three Web Attacks under a new category of "Web Attack" and "FTP-Patator" and "SSH-Patator" under a new category called "Brute Force". We note that two labels "Infiltration" and "Heartbleed" have too few instances i.e., 36 and 11 respectively, as compared to other labels. In comparison to Benign, these are 0.001% and 0.0004%. At such a small percentage, we posit that there are too few instances for the proposed model to learn meaningful and equitable representation of these attacks. Therefore, "Infiltration" and "Heartbleed" were not included in experimentation. The updated dataframe shape is (2830696, 79), with the label distribution as shown in Table 2.

## Data preprocessing

Data preprocessing is an integral part of artificial intelligence systems, and the steps taken in this study encompass several key procedures. Initially, the dataset was cleaned by removing columns that just have constant values for all instances since these do not carry useful information for the model development. The dataframe shape after this step is (2830696, 71). We dropped instance that have empty cells or missing data. This can happen when data is not

**Table 3. Summary of data partitioning and label distribution for binary classification.**

| Classes | Train | Test | Total |
|---|---|---|---|
| Benign | 1466539 | 297986 | 2095057 |
| Malicious | 628518 | 127708 | 425694 |

**Table 4. Summary of data partitioning label distribution for multi-label classification.**

| Classes | Train | Test | Total |
|---|---|---|---|
| Benign | 1466539 | 628518 | 2095057 |
| DoS Hulk | 89610 | 38404 | 128014 |
| PortScan | 63486 | 27208 | 90694 |
| Bot | 1364 | 584 | 1948 |
| Web Attack | 1500 | 643 | 2143 |
| Brute Force | 6405 | 2745 | 9150 |
| DDos | 135621 | 58124 | 193745 |

logged in correctly during experimentation. The dataframe shape after this step becomes (2827829, 71). It is possible to have a scenario where two or more instances have the exactly same features and labels i.e. duplicate instances. We remove such duplicates by keeping only the first occurrence, after which the dataframe shape becomes (2520751, 71). The label distribution for multilabel classification contain Benign (2095057) and other attack label group together in to new category called Malicious so that can perform binary classification. Therefore DoS (128014), PortScan (90694), Bot (1948), Web Attack (2143),Brute Force (9150), and DoS (193745) grouped in to a Malicious (425694) class. The train-test splits are generated with a 70–30 ratio with a fixed seed of "0" in a stratified manner across the two partitions. The label distribution and data partitioning of the dataset for the binary classification and multiclassification is provided in Tables 3 and 4 respectively.

## Feature selection and dimensionlity reduction

In the first stage, CFS is applied to the input data before feeding it into the neural network based classification model. The CFS technique chooses the features from a given subset based on the intercorrelation between features. Calculates collinearity for numerical values by using Pearson's correlation. To find individualized scores using one-hot encoding. The score can be calculated as

$$S = \frac{n_p a_{cf}}{\sqrt{n_p + n_p(n_p - 1)a_f f}} \tag{1}$$

The value or score denoted by S, np is the number of predictors for a given subset, and the average absolute intercorrelation between predictors and final features represented by acf. aff is the absolute intercorrelation between predictors. The primary goal is to reduce the dimensionality of the feature matrix by selecting the most relevant features while discarding less informative ones. The aim was to cause effect of simplifying the model and potentially improving its performance. Initially, we computed the correlation matrix and removed features that had a correlation of 95% or more; removing these changes the data frame shape from (2520751, 70) to (2520751, 47). After that, implementing feature engineering with a correlation factor of 90% correlation changes the data frame shape from (2520751, 70) to

(2520751, 39). We apply PCA with the 95% feature variance factor to optimize and reduce the model's dimension. Feature's dimension was decreased from 70 to 25 again. We use PCA, but now, this time with the factor of 90% feature variance; as a result, the dimension of the model was reduced from 70 to 20. In the reflection of reducing the dimension of the model, the complexity and size of the model reduced and improved the detection speed.

## Fully connected neural network(FCNN)

The second stage of our algorithm involves leveraging deep learning techniques to further optimize the model's performance. After the feature selection and dimensionality reduction stage, the processed data (with reduced features) is fed into the FCNN models. The models can learn complex patterns and representations from the data, potentially enhancing performance. The connections between the layers in FCNNs are typically one way in the forward direction and flow data continuously from the input layer to the output layer, as shown in Fig 3.

The fully connected neural network (FCNN) consists of a set of neurons in a multilayer network that are densely (fully) connected with each other. In such a setup, every neuron in the first layer receives as input the value of each attribute of the feature matrix and returns a single output. Consequently, each neuron in the second receives as input the output from all neurons in the preceding layer. At its core, it performs a dot product of all the input values along with the weights for obtaining the output followed by a non-linear function that provides a decision regarding the output value. Each hidden layer applies an activation function before creating an

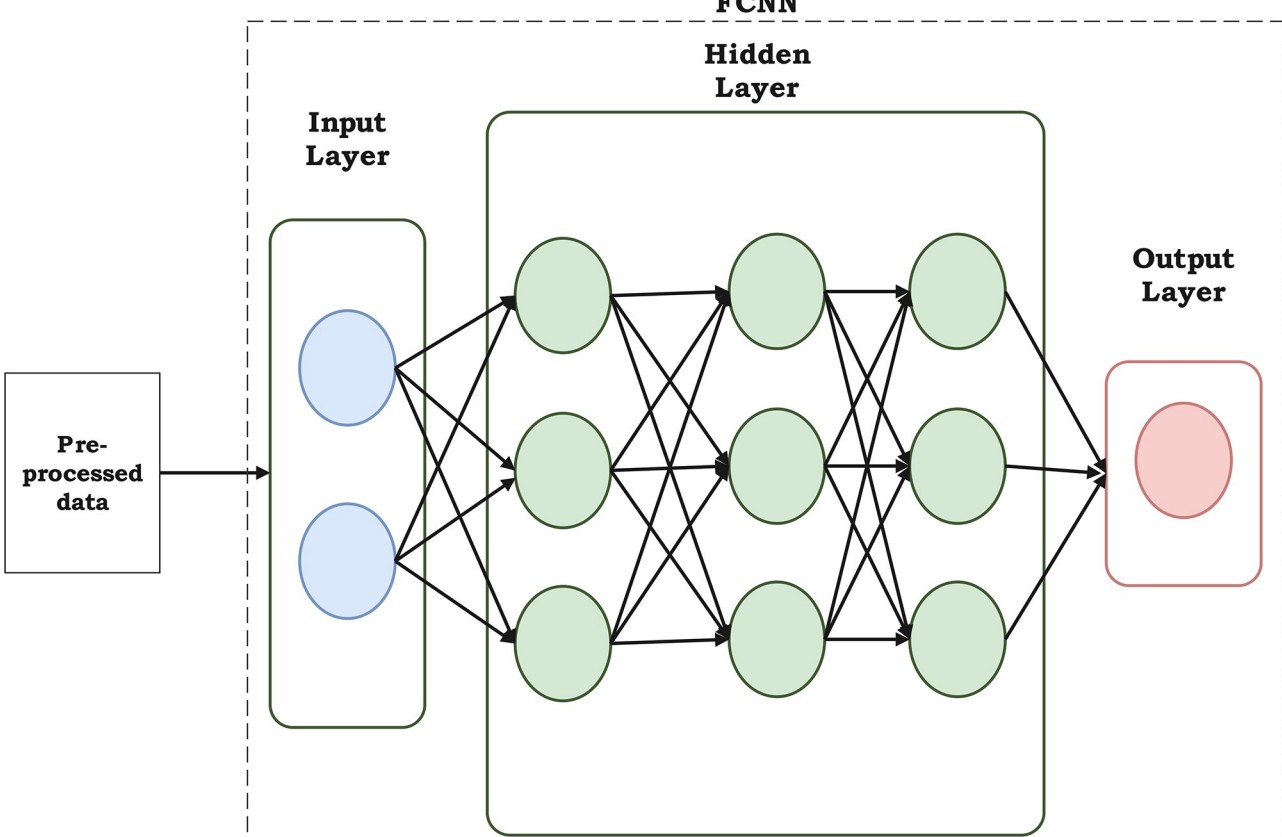

**Fig 3. Fully connected neural network.**

output, which is necessary to improve learning and approximation to avoid non-linear nature of real-world issue. By using Dropout over-fitting problem reduced in neural network. It seems sense that the primary function of the dropout layer is to eliminate any noise that might be present in the input of the neurons. As a result, overfitting of the model is avoided. Softmax is used for the output layer. Softmax activation function is used which is given in following equation

$$x_i = \frac{e^{a_i}}{\sum_{j=1}^{n} e^{a_j}} \tag{2}$$

aj represent the score associated with class(i). In this study Adam optimizer used and apply a sparse categorically cross-entropy loss function to adjust the optimizer weights. To calculate the binary cross entropy loss function, which measures a sample loss

$$Loss(x, \hat{x}) = -\frac{1}{n} \sum_{i=1}^{n} x_i \log(\hat{x}_i) + (1 - x_i) \log(1 - \hat{x}_i) \tag{3}$$

The multi class cross-entropy loss function is utilized to measure the loss of a sample by the following equation

$$Loss_{(x, \hat{x})} = -\sum_{i=1}^{n} x_i \log(\hat{x}_i) \tag{4}$$

In this paper, we design and develop 64D, 32D, 16D, and 2D dense layers of FCNN for anomaly detection in binary classification for autonomous vehicular networks

## Post training model quantization

The model size is reduced by post-training quantization in which weights are represented in terms of 16-bit floating point or 8-bit unsigned integer, as opposed to 32-bit floating point. The following equation depicts the data quantization procedure. The values to be mapped and the mapped attribute values are represented by the letters B and b respectively. The min and max represents minimum and maximum values for every packet.

$$B = \left(\frac{b - \min}{\max - \min}\right) \times 255 \tag{5}$$

Deep learning models' storage needs can be drastically reduced using Post training quantization to represent numerical quantities with less precision. Since decreased precision operations use fewer computational resources and execute more quickly. The hyperparameter settings used in this study include a batch size, learning rate, the optimizer, and number of epochs detailing these and other parameters is presented in Table 5.

## Experimental results

The experimental setting for this study is shown in Table 6.

The model is validated in five settings, i.e., without feature selection, with CFS at 95% and CFS at 90% correlation factor, and PCA with 95% and 90% variance, to compare and determine the impact of the feature selection and dimensionality reduction techniques on the classification performance, parameter count, and the size of the trained model. We surmise that such techniques will reduce the parameter count and size since the dimensionality of the feature matrix is reduced. However, such definitive conclusions require data-driven analysis, which is the goal of this study. We utilize FCNN as the core classification model, starting with

**Table 5. Summary of hyperparameters.**

| Hyperparameter | Description |
|---|---|
| Number of Dense Layers | 4 |
| Neurons per layer | 64 neurons in the first layer, 32 neurons in the second, 16 in the third, and 2 neurons in the output layer for binary classification |
| Learning rate | 0.0001 |
| Batch size | 32 |
| Max. Epochs | 30 |
| EarlyStopping | Yes, patience of 10 epochs |
| optimizer | Adam |
| Activation function | ReLU (L1-L3), Softmax (L4) |
| Classification Metric | Macro-Averaged F1-score |

a dense layer of 64 neurons before reducing it to two neurons (due to binary classification). The aim was to train the model for 30 epochs. However, an early stopping mechanism was used to optimize the training process, where model training was stopped if performance improvement was not observed over five epochs. As per convention, a batch size of 32 was used for the model training process. The Adam optimizer [31] is used to train and optimize the weights of the FCNN model through gradient descent whilst using the "momentum" aspect. This method is the de facto standard for model training since it offers a trade-off between performance and computational requirements.

In this study, the model parameters were optimized using the Adam optimizer [32] with an initial learning rate of 0.01, which was reduced with a factor of 0.2 after patience of 5 epochs to a minimum learning rate of 0.0001 if no improvement is observed in the validation partition loss. Performance is measured in terms of percentage of macro-averaged F1-score, rounded-off to two significant digits after the decimal place. Initial experiment showed that F1-score increased when the learning rate is decreased from 0.001 to 0.0001, however, further reduction beyond 0.0001 did not yield improvement rather required more epochs to reach adequate classification performance. This data driven experimentation guided us to select 0.0001 as the minimum learning rate for the Adam optimizer.

The loss function plays a pivotal role in neural networks [33]. It is responsible for computing gradients, which, in turn, guide the adjustment of neural network biases and increasing and decreasing weights [34]. To this end, the CategoricalCross entropy loss function was to train the model parameters. In [35], classification accuracy was chosen as the performance measurement metric for deep learning models. However, given the label imbalance i.e. 10% 90%, using accuracy as the metric cannot yield adequate performance measurement as it is biased in favour of the majority class. This work uses macro-averaged f1-score as the metric, which is a fair metric to both classes. In this research, binary and multi-label classification

**Table 6. Experimental environment.**

| Model | Parameters |
|---|---|
| Processor | AMD Ryzen 9 PRO processor |
| Main frequency | 3.3GHz |
| RAM | 64GB |
| Operating System | Windows 11Pro |
| Experimental tool | Python 3.7 and TensorFlow2.2.0 |

## Model's performance using binary classification

The F1-score and loss of the FCNN model were evaluated during each epoch for both the training and validation sets. This assessment enables us to learn the model's ability to effectively distinguish between different types of traffic and determine the extent to which instances from the validation set are correctly classified. We trained FCNN model under five (05) experiemental settings: without feature selection, with CFS correlation (factor of 95% and 90%) and PCA with variance of 95% and 90%. The loss function and F1 score plot without using feature selection as seen in Fig 4. The average F1-score was 96% for training and validation partitions based on the CICIDS2017 dataset. Initial experimetation showed that the F1-score did not improve beyond 40 epochs, therefore, 40 was taken as the maximum number of epochs to train the model. Further, the actual number of epochs was tuned using the EarlyStopping criteria with a patience of 5 epochs to alleviate overfitting the training partition. Fig 5 show the training and validation graphs in terms of F1-score and loss for the five experiments.

A concise summary of observations is provided in Table 7.

The Table 7 shows that the transition from float32 to float16 and int8 quantization significantly reduces model size, making it more memory and energy efficient. Feature selection can be used for both performance improvement and model optimization in terms of parameter count reduction. When CFS90 is used (without quantization), the classification performance in terms of f1score improves from 96.48% to 98.43% whilst reduces the model parameter count from 7192 to 5208. When CFS90 is used under float16 setting, the model parameters are reduced from 7192 to 5208 and due to 16-bit floating point quantization, the model size is reduced by factor of (28.09–9)/28.09 i.e. 67.96% while marginally improving the F1-score from 94.48% to 94.84%. The use of PCA90 technique reduces the model parameters as well as the size by a factor of (7192–3992)/7192 i.e. 44.49% whilst have a detriment on classification performance by an amount of 0.26% under float32 setting. It is interesting to note that the Tflite format has a slightly larger file size for float32 as compared to the hdf5 format, we posit that extra information is stored within the model parameters.

## Model's performance for multi-label classification

Table 8 provides a summary of multi-label classification results with various feature selection and model quantization settings. In the best case scenario, the model is optimized in terms of parameter count (7327 to 4447) and model size (28.62 KB to 8 KB) while retaining the same classification performance as the default setting (No FS and 32-bit float). This is achieved by PCA95 setting for FS with float16 quantization. We also note that PCA90 can offer a slight improvement in the classification performance i.e. 68.87% to 69.22% while reducing the parameter count from 7327 to 4127 as well as reducing the model size by (28.62–16.12)/28.62 i.e. 43.68%. Our experiment results also suggest that the process of model optimization is a data-driven task and thorough investigation is required before a design is finalized. For example, consider the setting of PCA90 with 8-bit quantization, where model performance crashes to 55.71% for multilabel classification task while reducing the model parameter count as well as the size by a significant margin. Moreover, we did not find significant improvement in model size reduction from float16 to int8 quantization. This is because the TensorFlow Model Optimization (tfmot) toolkit retains certain parameters in float16 format and does not convert them to int8 as it seeks to alleviate performance degradation.

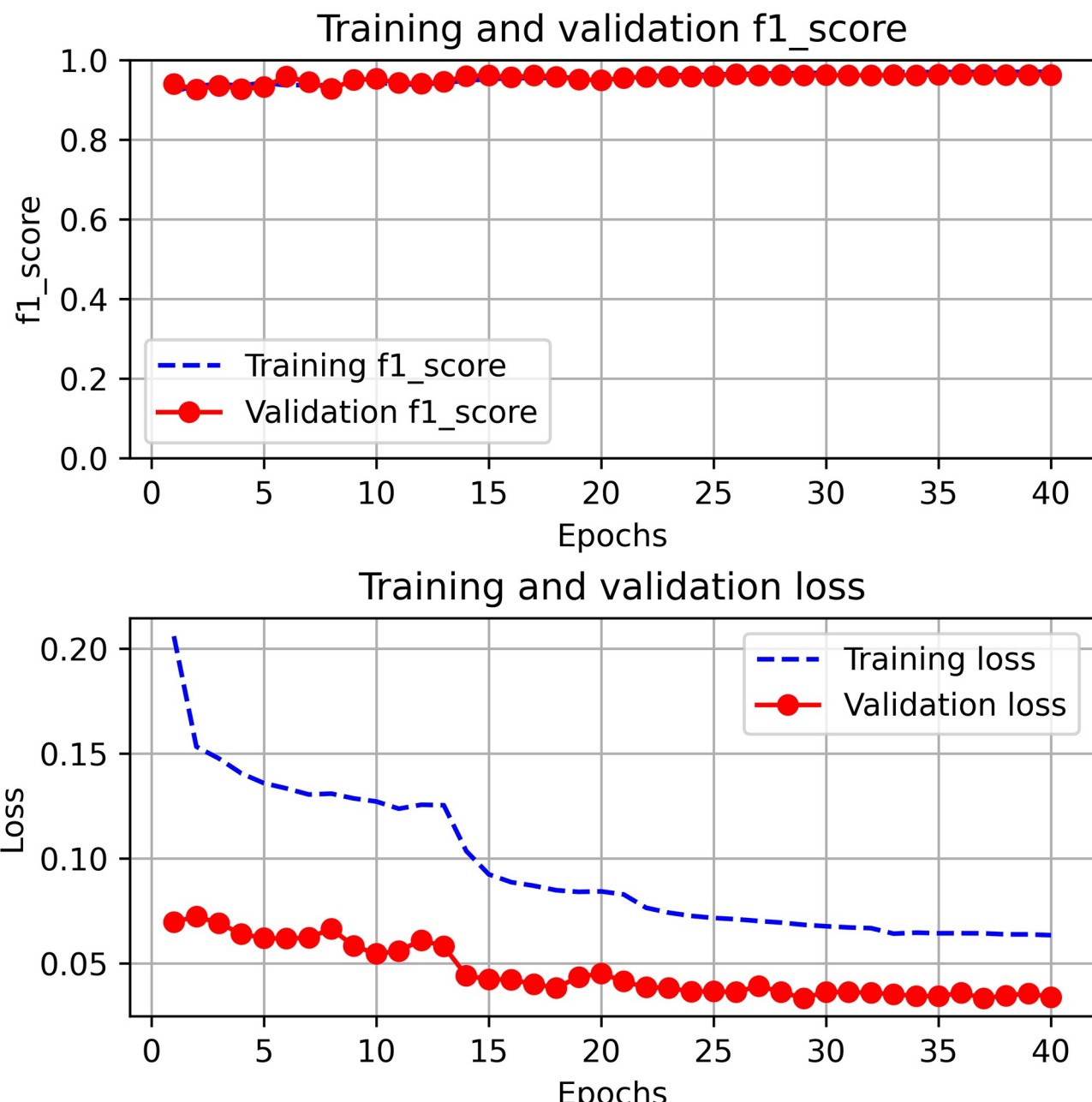

**Fig 4. Loss function and F1 score without feature selection.**

## Discussion

In this manuscript, we proposed a hybrid approach for automated intrusion detection for the VANET scenario. The hybrid approach, that involves the use of feature selection and dimensionality reduction of the feature matrix followed by post-training model quantization, seeks to yield optimized machine learning models for the task of intrusion detection. The model's complexity, the feature matrix's size, and the optimization methods used can all substantially impact the time it takes to train a DL model. There are various benefits to training

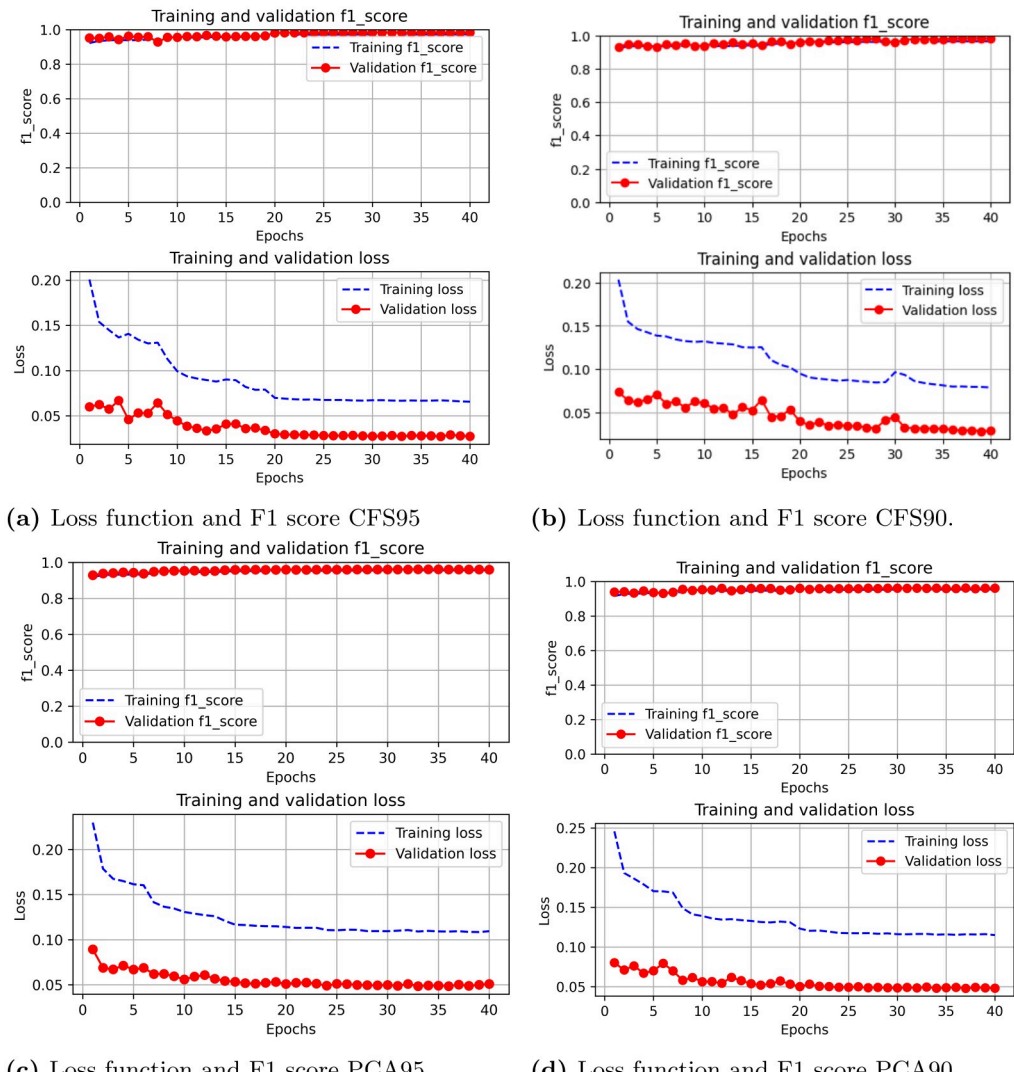

**Fig 5. Training and validation graphs in terms of F1-score and loss for the five experiments.** (**a**) Loss function and F1 score CFS95. (**b**) Loss function and F1 score CFS90. (**c**) Loss function and F1 score PCA95. (**d**) Loss function and F1 score PCA90.

the suggested model on a graphics processing unit (GPUs) rather than a central processing unit [36].

## Speed

The GPUs are made to execute the kinds of calculations needed for deep learning models, which is one of the key benefits of training a hybrid IDS solution [37]. In [38] author used the CICIDS2017 dataset combined with CNN and GRU techniques to optimize network parameters for Intrusion Detection Systems (IDS) and achieved an accuracy of 98.73%. However, given the classification imbalance, accuracy was not the most suitable metric. In [39], the authors compared three DNN, LSTM, and CNN deep learning models for IoT Intrusion Detection Systems. The study emphasizes the superiority of deep learning models for IoT IDS

**Table 7. Summary of results for binary classification performance and model quantization for various feature engineering and dimensionality reduction experiments.**

| Details | without FS | CFS95 | CFS90 | PCA95 | PCA90 |
|---|---|---|---|---|---|
| Float32(hdf5) | | | | | |
| Parameters | 7192 | 5720 | 5208 | 4312 | 3992 |
| Size | 28.09KB | 22.34KB | 20.34KB | 16.84KB | 15.59KB |
| F1 score | 96.48% | 98.42% | 98.43% | 96.17% | 96.22% |
| Float32(tflite) | | | | | |
| Size | 31KB | 25KB | 23KB | 20KB | 18KB |
| F1 score | 96.48% | 98.42% | 98.43% | 96.17% | 96.22% |
| Float16(tflite) | | | | | |
| Size | 11KB | 10KB | 9KB | 9KB | 8KB |
| F1 score | 94.94% | 93.85% | 94.84% | 94.21% | 93.41% |
| Int8(tflite) | | | | | |
| Size | 11KB | 9KB | 9KB | 9KB | 8KB |
| F1 score | 93.34% | 90.04% | 93.54% | 91.09% | 90.51% |

to enhance the accuracy in attack prediction. The author achieved 94.61% accuracy in DNN, while LSTM and CNN achieved 97.67% and 98.61%, respectively. The existing study did not focus on model optimization. whereas one can compare in terms of classification performance, a comparison is not possible in terms of model optimization. Training the suggested model, as a result, are significantly faster because of the model's drastically reduced size [40]. The investigation suggests that correlation-based feature selection provides better classification results than PCA-based dimensionality reduction for both binary as well as multi-label classification setup, and therefore, should be preferred. In experiments for model quantization that explored the quantization of model parameter values from 32-bit floating point to 16-bit floating point and 8-bit unsigned values, it was identified that the classification performance suffers a slight decrease, however, we posit that this is an acceptable trade-off given that the decrease in model size and more efficient processing provides the benefit of deployment in resource-constrained environments such as autonomous vehicular networks. Overall, we find float16 provides the

**Table 8. Summary of results for multi-label classification performance and model quantization for various feature engineering and dimensionality reduction experiments.**

| Details | without FS | CFS95 | CFS90 | PCA95 | PCA90 |
|---|---|---|---|---|---|
| Float32(hdf5) | | | | | |
| Parameters | 7327 | 5855 | 5343 | 4447 | 4127 |
| Size | 28.62KB | 22.87KB | 20.87KB | 17.37KB | 16.12KB |
| F1 score | 68.87% | 68.01% | 67.55% | 68.38% | 65.49% |
| Float32(tflite) | | | | | |
| Size | 31KB | 26KB | 24KB | 20KB | 19KB |
| F1 score | 68.87% | 68.01% | 67.55% | 68.38% | 96.22% |
| Float16(tflite) | | | | | |
| Size | 11KB | 10KB | 9KB | 8KB | 8KB |
| F1 score | 62.42% | 64.32% | 66.97% | 68.93% | 59.61% |
| Int8(tflite) | | | | | |
| Size | 11KB | 9KB | 9KB | 9KB | 8KB |
| F1 score | 64.04% | 56.63% | 59.41% | 64.10% | 55.71% |

best tradeoff as compared to float32 and int8 quantization. The current work lays down the foundation for future research directions that may explore additional feature selection methods and further investigate the trade-offs between model size and classification performance with different quantization techniques as well as experiments on embedded edge-ML devices considering real-world deployment scenarios and computational constraints.

## Conclusion and future work

This study presents a hybrid approach for intrusion detection in vehicular networks that effectively balances classification performance and model efficiency. Our method, which combines automated feature engineering through correlation-based feature selection (CFS) and principal component analysis (PCA) with optimized deep learning techniques, has yielded promising results. The proposed hybrid model significantly improved classification performance, increasing the F1-score from 96.48% to 98.43%. Moreover, our feature engineering techniques reduced the model size from 28.09 KB to 20.34 KB, optimizing both performance and resource usage. Post-training quantization further compressed the model to 9 KB, demonstrating substantial optimization potential. Experiments using the CICIDS 2017 dataset validated the model's effectiveness in classifying malicious traffic in vehicular network scenarios. These findings prove that it is possible to develop high-performing intrusion detection systems with low computational complexity, making them suitable for deployment in resource-constrained vehicular ad-hoc network (VANET) environments. Overall, this research aimed to advance the development of efficient and reliable intrusion detection systems tailored for VANETs and other resource-constrained real-world deployments

In the current work, we used post-training model weights quantization as the approach for optimizing the model size. However, using this study as the base, we propose to explore techniques such as parameter pruning, weight clustering, and quantization-aware training for model optimization and compare their performance in terms of classification and model size reduction.

## Supporting information

**S1 Appendix.**
(PDF)

## Author Contributions

**Conceptualization:** Fayaz Hassan, Zafi Sherhan Syed.

**Data curation:** Fayaz Hassan, Nadeem Ahmed, Mana Saleh Al Reshan, Yousef Asiri.

**Formal analysis:** Aftab Ahmed Memon, Saad Said Alqahtany, Nadeem Ahmed, Yousef Asiri.

**Investigation:** Zafi Sherhan Syed, Aftab Ahmed Memon, Mana Saleh Al Reshan, Yousef Asiri, Asadullah Shaikh.

**Methodology:** Fayaz Hassan, Zafi Sherhan Syed, Mana Saleh Al Reshan, Yousef Asiri.

**Resources:** Zafi Sherhan Syed, Aftab Ahmed Memon, Saad Said Alqahtany, Nadeem Ahmed, Mana Saleh Al Reshan.

**Software:** Saad Said Alqahtany.

**Supervision:** Asadullah Shaikh.

**Validation:** Fayaz Hassan, Zafi Sherhan Syed, Aftab Ahmed Memon, Saad Said Alqahtany, Asadullah Shaikh.

**Visualization:** Nadeem Ahmed, Mana Saleh Al Reshan, Asadullah Shaikh.

**Writing – original draft:** Fayaz Hassan, Zafi Sherhan Syed.

**Writing – review & editing:** Fayaz Hassan, Zafi Sherhan Syed, Aftab Ahmed Memon, Saad Said Alqahtany, Nadeem Ahmed, Mana Saleh Al Reshan, Yousef Asiri, Asadullah Shaikh.

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
