## [Decision Letter · Decision Letter 0]

28 Aug 2024

PONE-D-24-33805A Hybrid Approach for Intrusion Detection in Vehicular Networks using Feature Selection and Dimensionality Reduction with Optimized Deep LearningPLOS ONE

Dear Dr. Shaikh,

Thank you for submitting your manuscript to PLOS ONE. After careful consideration, we feel that it has merit but does not fully meet PLOS ONE’s publication criteria as it currently stands. Therefore, we invite you to submit a revised version of the manuscript that addresses the points raised during the review process.

We look forward to receiving your revised manuscript.

Kind regards,

Zhiquan Liu, Ph.D.

Academic Editor

PLOS ONE

**Journal Requirements:**

**Additional Editor Comments:**

Both the reviewers hold a positive attitude towards this paper and point out some existing problems in this paper. The author is requested to make careful revisions according to the review comments, and then submit its revised version.

Reviewers' comments:

Reviewer's Responses to Questions

**Comments to the Author**

1. Is the manuscript technically sound, and do the data support the conclusions?

Reviewer #1: Yes

Reviewer #2: Partly

2. Has the statistical analysis been performed appropriately and rigorously? 

Reviewer #1: Yes

Reviewer #2: Yes

3. Have the authors made all data underlying the findings in their manuscript fully available?

Reviewer #1: Yes

Reviewer #2: Yes

4. Is the manuscript presented in an intelligible fashion and written in standard English?

Reviewer #1: Yes

Reviewer #2: Yes

5. Review Comments to the Author

**Reviewer #1: **Reviewer Comments on a Hybrid Approach for Intrusion Detection in Vehicular Networks using Feature Selection and Dimensionality Reduction with Optimized Deep Learning

1. The motivation and contribution of this paper should be stated more clearly in the abstract to better understand from the beginning of the study. Authors are advised to be precise in the abstract, and structure your abstract as follows- 1) Background 2) Aim/Objective 3) Methodology 4) Results 5) Conclusion. Write 2-4 lines for each and merge everything in one paragraph (200-300 Words) without any subheading.

2. The introduction section should be improved by adding more recent works in this area and providing a more accurate and informative.

3. Authors should justify how the dataset CIC-IDS2017 is relevant datasets that replicated the network traffic in the vehicular networks.

4. The research does not explicitly address the issue of class imbalance in the training data. Imbalanced datasets can lead to biased models, and it's important to understand how the proposed research handles this issue.

5. The related work section is very small, an updated and complete literature review should be conducted. Some latest papers listed below which studied similar problems can be discussed.

https://www.sciencedirect.com/science/article/abs/pii/S1568494624002084

https://link.springer.com/article/10.1007/s10207-023-00803-x

https://www.frontiersin.org/articles/10.3389/fcomp.2023.997159/full

6. In the end of related work section, highlight in 10-15 lines what overall technical gaps are observed in existing techniques that led to the design of the proposed research

7. The hyperparameters of the fully connected neural network should be included in a Table format with all the parameters and values stated.

8. The hyperparameters of the fully connected neural network should be included in a Table format with all the parameters and values stated.

9. The number of features and list of features selected by CFS and PCA should be included and put in a Table format.

10. The authors should provide a flowchart and pseudocode for the algorithms used.

11. Authors should compare the results obtained with the existing studies in the literature.

12. Refine the conclusion to succinctly summarize the key findings and contributions.

13. Although the English is generally quite good, there are quite a few minor grammatical errors, and a careful read-through is needed to eliminate these errors. The spelling mistake should be corrected by reading through the manuscript.

**Reviewer #2: **This manuscript proposed a low-complexity model that can be used for cyberattack detection in VANET scenario. This is a meaningful work. Following are some suggestions to the authors to improve the quality of the paper. The figures can be refined to make the content clearer. The proposed scheme is not detailed enough. The organization of chapters should be refined. Please specify the advantages of the proposed solution in the abstract, preferably in a specific numerical form. The discussion or comparisons with more recent related schemes, such as trove: a context awareness trust model for vanets using reinforcement learning, and ppru: a privacy-preserving reputation updating scheme for cloud-assisted vehicular networks are suggested. More background information, especially the real-world application scenarios and specific challenges faced in vehicular collaborative systems, should be provided. The authors should discuss future work from the perspective of the current model's existing problems.

6. PLOS authors have the option to publish the peer review history of their article (what does this mean?). If published, this will include your full peer review and any attached files.

Reviewer #1: **Yes: **Yakub Kayode Saheed

Reviewer #2: No

---

## [Author Response · Author response to Decision Letter 0]

10 Oct 2024

Reviewer 1

Dear Reviewer,

Thank you very much for your comments. We appreciate your opinion and recommendations. Considering your suggestions, we made following modification in our revised version.

Concern # 1: 1. The motivation and contribution of this paper should be stated more clearly in the abstract to better understand from the beginning of the study. Authors are advised to be precise in the abstract, and structure your abstract as follows- 1) Background 2) Aim/Objective 3) Methodology 4) Results 5) Conclusion. Write 2-4 lines for each and merge everything in one paragraph (200-300 Words) without any subheading

Author Response: Thank you for the constructive feedback. We have updated the abstract to better describe our work. The revised abstract has been precise as per reviewer’s suggestion

Concern # 2: The introduction section should be improved by adding more recent works in this area and providing a more accurate and informative

Author response: Thank you for providing a valuable suggestion by adding the latest related work, our manuscript has greatly improved. We updated the manuscript, and the more recent works added in the Introduction section. We added following related work with following paragraphs:

Title: [XAI-ADS An Explainable Artificial Intelligence Framework for Enhancing Anomaly Detection in Autonomous Driving Systems]

“Vehicular networks have the potential to transform transportation systems by enabling advanced communication between vehicles and infrastructure. Many research studies[ref] have focused on enhancing the security of vehicular networks.”

Title: [Smart Collaborative Intrusion Detection System for Securing Vehicular Networks Using Ensemble Machine Learning Model]

“The rapid growth of vehicular network systems has revealed significant vulnerabilities, emphasizing the need for smart detection systems specifically designed for vehicular networks. [ref]”

Concern # 3: Authors should justify how the dataset CIC-IDS2017 is relevant datasets that replicated the network traffic in the vehicular networks.

Author response: While the dataset was not specifically designed for vehicular ad hoc networks (VANETs), many of its features can still be relevant for analyzing network traffic in such environments. It should be noted that there do not exist any bespoke dataset for VANET IDS, and we used CICIDS2017 for task at hand.

VANETs needs specific characteristics to secure communication, the presence of wireless networks, and the potential for malicious attacks. Therefore, our rationale for utilizing the CICIDS dataset in our research is that it serves as the starting point and proof of concept for assessing the efficacy of our proposed IDS in a real-world context

In the context of vehicular networks, the following attributes from CICIDS2017 are relevant for detecting malicious traffic.

Flow Bytes per Second: This measures the data transfer rate of a communication flow, which can be useful for detecting flooding attacks in VANETs.

Packet Length (Mean, Std, Max, Min): These features describe the size of data packets in the network. Unusual packet sizes could signal malicious intent.

 Flow Packets per Second: High-frequency packet flows may indicate malicious scanning or DDoS activity in vehicular networks.

 Idle Time: The amount of idle time in communication. In VANETs, long idle times might indicate jamming or disruption attacks.

Active Time: Time a flow remains active can be relevant in detecting anomalies in expected vehicular communication patterns.

 PSH Flags Count, URG Flags Count: These flags in the TCP protocol can indicate urgency in communication, but attackers sometimes misuse these flags to bypass IDS systems.

Number of Failed Login Attempts: This can indicate brute-force attempts, which might target the vehicular control systems or connected infrastructure.

Destination IP, Source IP: Monitoring IP addresses in a vehicular network can help identify unauthorized or malicious nodes trying to communicate with the system.

Packet Loss Rate: High packet loss could be a sign of a denial of service (DoS) attack or network congestion, which may compromise vehicular communication systems.

Concern # 4: The research does not explicitly address the issue of class imbalance in the training data. Imbalanced datasets can lead to biased models, and it's important to understand how the proposed research handles this issue

Author response: We acknowledge and agree with the reviewer that class imbalance is a serious matter of concern for machine learning in general and IDS classification in particular. As a result, we included the following measure in the ML pipeline, including defining a “class_weight” in the model training process i.e. within the model.fit parameter, we passed the proportional weight of each class. Furthermore, the performance of the models was evaluated using macro-average f1-score, which is robust in imbalanced learning scenarios. Relevant information was included in Sections “Dataset” and “Experimental Results”. This have been further clarified in the updated manuscript.

Concern # 5: The related work section is very small, an updated and complete literature review should be conducted. Some latest papers listed below which studied similar problems can be discussed.

https://www.sciencedirect.com/science/article/abs/pii/S1568494624002084

[Author et al.] proposed the Modified Genetic Algorithm (MGA) and a Long Short-Term Memory (LSTM) network to detect cyberattacks in IoT networks, particularly in smart city environments. Their approach focuses on edge computing in centralized IDS , addressing challenges in real-time detection and feature optimization.

https://link.springer.com/article/10.1007/s10207-023-00803-x

https://www.frontiersin.org/articles/10.3389/fcomp.2023.997159/full

Author response: We updated the manuscript, and the more recent work added in the Related work section. We added following added related work with following paragraphs:

 Title: [Modified genetic algorithm and fine-tuned long short-term memory network for intrusion detection in the internet of things networks with edge capabilities]

 “proposed the Modified Genetic Algorithm (MGA) and a Long Short-Term Memory (LSTM) network to detect cyberattacks in IoT networks, particularly in smart city environments. Their approach focuses on edge computing in centralized IDS , addressing challenges in real-time detection and feature optimization. “

Title: [A voting gray wolf optimizer-based ensemble learning models for intrusion detection in the Internet of Things]

“In {ref}, ensemble learning techniques Voting Gray Wolf Optimizer (GWO) used to enhance the accuracy of Intrusion Detection Systems (IDS) for IoT networks.”

Title: [A novel hybrid autoencoder and modified particle swarm optimization feature selection for intrusion detection in the internet of things network]

“ Another study {ref} the author highlights the potential of hybrid techniques combining feature selection and optimization methods to improve IDS effectiveness”.

Concern # 6 In the end of related work section, highlight in 10-15 lines what overall technical gaps are observed in existing techniques that led to the design of the proposed research

Author response: Thank you very much for your valuable suggestions. We added the following paragraph in the related work section.

1. One finds that prior works use 'accuracy' as classification performance metric even though datasets have an imbalanced class distribution, which means that performance is always optimistic in favor of the majority class that is often the non-malicious normal traffic. This is a major weakness as it means that the results do not show the correct picture of the IDS performance.

2. Model optimization, in terms of classification performance and size, is not adequately addressed in existing research literature. One notes that in a majority of cases, model optimization is limited in scope to increasing classification performance and does not address reducing model size or reducing the number of parameters through feature engineering.

3. Our literature survey does not find prior work that studies the impact of using quantization for model size reduction on the classification performance for VANETs.

The proposed addresses these three gaps in pursuit of developing a hybrid models for IDS for scenario.

Concern # 7,8 The hyperparameters of the fully connected neural network should be included in a Table format with all the parameters and values stated

Author response: The concern is addressed. The hyperparameter Table have been added in the section “Proposed System Methodology”. Table: 5 added in the revised manuscript

Concern # 9 The number of features and list of features selected by CFS and PCA should be included and put in a Table format

Author response: We have number of features and list of features selected by CFS at the end of the manuscript in a newly created section titled Appendix. However, PCA is a dimensionality reduction technique that operates by transforming the original features into a new set of principal components. Since PCA combines and transforms the feature space, it does not select specific features in the same way CFS does. Therefore, we cannot identify features selected by PCA during experiment.________________________________________

Concern # 10 The authors should provide a flowchart and pseudocode for the algorithms used. 

Author response: We have added the pseudocode at the end of the manuscript in a newly created section titled Appendix. The flowchart is provided in the section Proposed System Methodology with the title of figure Proposed framework.

Concern # 11 Authors should compare the results obtained with the existing studies in the literature 

Author response: The concern is addressed. We compared the existing study in our Discussion section. The existing study did not focused on model optimization. whereas one can compare in terms of classification performance, a comparison is not possible in terms of model optimization

[Composition of Hybrid Deep Learning Model and Feature Optimization for Intrusion Detection System] 

“In[ref] author used the CICIDS2017 dataset combined with CNN and GRU techniques to optimize network parameters for Intrusion Detection Systems (IDS) and achieved an accuracy of 98.73%. However, given the classification imbalance, accuracy was not the most suitable metric.”

[Deep learning algorithms for intrusion detection systems in internet of things using CIC-IDS 2017 dataset]

“In [ref], the authors compared three DNN, LSTM, and CNN deep learning models for IoT Intrusion Detection Systems. The study emphasizes the superiority of deep learning models for IoT IDS to enhance the accuracy in attack prediction. Author achieved 94.61% accuracy in DNN, while LSTM and CNN achieved 97.67% and 98.61%, respectively.” 

Concern # 12 Refine the conclusion to succinctly summarize the key findings and contributions.

Author response: Thank you very much for your valuable suggestions. We updated the conclusion by adding following paragraph.

“This study presents a hybrid approach for intrusion detection in vehicular networks that effectively balances classification performance and model efficiency. Our method, which combines automated feature engineering through correlation-based feature selection (CFS) and principal component analysis (PCA) with optimized deep learning techniques, has yielded promising results. The proposed hybrid model significantly improved classification performance, increasing the F1-score from 96.48% to 98.43%. Moreover, our feature engineering techniques reduced the model size from 28.09 KB to 20.34 KB, optimizing both performance and resource usage. Post-training quantization further compressed the model to 9 KB, demonstrating substantial optimization potential. Experiments using the CICIDS 2017 dataset validated the model's effectiveness in classifying malicious traffic in vehicular network scenarios. These findings prove that it is possible to develop high-performing intrusion detection systems with low computational complexity, making them suitable for deployment in resource-constrained vehicular ad-hoc network (VANET) environments.”

Concern # 13 Although the English is generally quite good, there are quite a few minor grammatical errors, and a careful read-through is needed to eliminate these errors. The spelling mistake should be corrected by reading through the manuscript

Author response: Thank you very much for letting us know about the language in our manuscript We carefully revised manuscript and improved the grammatical errors.

Reviewer 2

Dear Reviewer,

We are really very grateful for your valuable suggestions. We observed that your suggestions/ Comments have improved overall manuscript. We carefully considered every single comment and incorporated it in our revised manuscript. Following are the comments and their responses. 

Concern # 1: The figures can be refined to make the content clearer

Author response: Thank you for gaining our attention regarding the quality of the figures. Keeping in mind the suggestion of the Reviewer, We have enhanced the DPI and text size for figures 1, 2 and 3.________________________________________

Concern # 2: The proposed scheme is not detailed enough.

Author response: Thank you for feedback regarding the level of detail in our proposed scheme. we have updated the manuscript by including pseudocode for the algorithms (section Appendix) and a table outlining the hyperparameters (in Table:5) and the features selected by CFS. These additions aim to clarify the proposed method and improve the overall comprehensiveness of the manuscript.

Concern # 3: The organization of chapters should be refined.

Author response: Thank you very much for your constructive suggestion. We have updated the organization of section in revised manuscript. The following paragraph is updated version of the organization of section

This study is organized as follows: Section 2(Related Work) presents a review of recent advances in intrusion detection, Section 3 (Propose system methodology) summarizes the research gap and the problem statement followed by a detailed description of the proposed system methodology, including the dataset, data preprocessing steps, feature selection, dimensionality reduction, and the fully connected neural network (FCNN) architecture, as well as post-training model quantization techniques. Section 4 presents details experiment results, evaluating the model’s performance for both binary and multi-class classification tasks. Section 5 provides an in-depth discussion of the findings. Finally, conclusions and potential directions for future work are provided in Section 6.

Concern # 4: Please specify the advantages of the proposed solution in the abstract, preferably in a specific numerical form. 

Author Response: Thank you for the constructive feedback. We have updated the abstract to specify the advantages of the proposed solution in numerical form. We have added the following paragraph in the abstract.

“Results demonstrate the effectiveness of feature engineering which improves the classification f1score from 96.48% to 98.43%. It also reduces the model size from 28.09 KB to 20.34 KB thus optimizing the model in terms of both classification performance and model size. Post-training quantization further optimizes the model size to 9 KB.” 

Concern # 5: The discussion or comparisons with more recent related schemes, such as trove: a context awareness trust model for vanets using reinforcement learning, and ppru: a privacy-preserving reputation updating scheme for cloud-assisted vehicular networks are suggested. 

Author Response: Thank you for providing a valuable sugges

---

## [Editor Report · Decision Letter 1]

14 Oct 2024

A Hybrid Approach for Intrusion Detection in Vehicular Networks using Feature Selection and Dimensionality Reduction with Optimized Deep Learning

PONE-D-24-33805R1

Dear Dr. Shaikh,

We’re pleased to inform you that your manuscript has been judged scientifically suitable for publication and will be formally accepted for publication once it meets all outstanding technical requirements.

Kind regards,

Zhiquan Liu, Ph.D.

Academic Editor

PLOS ONE

Additional Editor Comments (optional):

Accept
---

## [Editor Report · Acceptance letter]

17 Oct 2024

PONE-D-24-33805R1 

PLOS ONE

Dear Dr. Shaikh, 

I'm pleased to inform you that your manuscript has been deemed suitable for publication in PLOS ONE. Congratulations! Your manuscript is now being handed over to our production team.

Kind regards, 

on behalf of

Professor Zhiquan Liu 

Academic Editor

PLOS ONE